# Reverse Engineering Cancer: Inferring Transcriptional Gene Signatures from Copy Number Aberrations with ICAro

**DOI:** 10.3390/cancers11020256

**Published:** 2019-02-22

**Authors:** Davide Angeli, Maurizio Fanciulli, Matteo Pallocca

**Affiliations:** 1Department of Paediatric Haematology, IRCCS Ospedale Pediatrico Bambino Gesù, 00146 Rome, Italy; davide.ang@gmail.com; 2SAFU Unit, IRCCS Regina Elena National Cancer Institute, 00144 Rome, Italy; maurizio.fanciulli@ifo.gov.it

**Keywords:** transcriptional signatures, copy number variation, copy number aberration, TCGA mining, cancer CRISPR, firehose, gene signature extraction, gene loss biomarkers, gene inactivation biomarkers, biomarker discovery

## Abstract

The characterization of a gene product function is a process that involves multiple laboratory techniques in order to silence the gene itself and to understand the resulting cellular phenotype via several omics profiling. When it comes to tumor cells, usually the translation process from in vitro characterization results to human validation is a difficult journey. Here, we present a simple algorithm to extract mRNA signatures from cancer datasets, where a particular gene has been deleted at the genomic level, ICAro. The process is implemented as a two-step workflow. The first one employs several filters in order to select the two patient subsets: the inactivated one, where the target gene is deleted, and the control one, where large genomic rearrangements should be absent. The second step performs a signature extraction via a Differential Expression analysis and a complementary Random Forest approach to provide an additional gene ranking in terms of information loss. We benchmarked the system robustness on a panel of genes frequently deleted in cancers, where we validated the downregulation of target genes and found a correlation with signatures extracted with the L1000 tool, outperforming random sampling for two out of six L1000 classes. Furthermore, we present a use case correlation with a published transcriptomic experiment. In conclusion, deciphering the complex interactions of the tumor environment is a challenge that requires the integration of several experimental techniques in order to create reproducible results. We implemented a tool which could be of use when trying to find mRNA signatures related to a gene loss event to better understand its function or for a gene-loss associated biomarker research.

## 1. Background

Translational research has been hard at work trying to find a way to characterize genes and gene product functions for decades. One successful approach is the study of particular contexts where the gene expression of interest is perturbed. In the past, biologists mostly tried to characterize gene functions by overexpressing its mRNA, whereas more recently, several tools have been introduced in the field of Cellular and Molecular Biology to erase a gene (or its mRNA). Furthermore, a rapid evolution of induced DNA/RNA ablation techniques have emerged from perfectible approaches including siRNA/shRNA to highly specific ones such as TALEN and CRISPRs/Cas9 [1,2].

An induced gene deletion (or mRNA ablation) event brings about a series of phenotypes, both as direct consequences of the gene/protein absence and as epiphenomena mediated by the cellular environment response of such a relevant change.

The granular study of these phenotypes has been accelerated dramatically by the introduction of omic technologies in basic and translational research. For instance, we can easily take a transcriptome-wide picture of the mRNA status or the profile of a large panel of metabolites. All these data can easily help the investigators to apply the “guilt by association” approach in order to better understand a gene function by looking at the correlated omic response [3]. In spite of the elegant workflow (perturbation → omics → understanding), the process is hindered by a series of issues.

In regards to silencing technologies, while CRISPRs have promised to lead much less off-target effects than shRNAs, they still are a challenging technique for several laboratories worldwide and even show little correlation with RNA interference screens, a worrying scenario since thousands of mechanistic papers on cellular and molecular biology are based on these tools [4]. Furthermore, most of these characterizations are conducted in vitro, where the reproducibility of results is being pointed out as a major issue [5,6,7].

Several efforts have been made towards also automating and standardizing in vitro results to make them reproducible. Among these proposals, the L1000 connectivity map [8,9] is a clear example of a thorough characterization of the mRNA response of thousands of compounds (shRNA, overexpression, and drugs) in several cell lines.

However, when the whole question shifts to a difficult cellular context such as cancer, the situation worsens. The network of intercellular and intracellular interactions of the tumor macroenvironment is extremely complex and inevitably fails to be modeled by a simple mono-population cell line. In relation to this, organoids are an interesting promise [10], but most medium- and small-sized laboratories worldwide still do not have access to these kinds of models.

On the other hand, one resource that is available to any oncology-based research group is access to public cancer datasets. Only The Cancer Genome Atlas (TCGA) contains several molecular profiles from more than 11,000 patients at the time of the writing [11]. We tried to reason whether we could extract huge amounts of data to make the process of elucidating gene functions in cancer contexts easier and more robust. For this reason, we implemented ICAro (gene signature Inference system from Copy number Aberrations), a framework that enables researchers to extract putative gene signatures from publicly available Cancer Genomic datasets.

This overall idea involves treating cancer as a Cas9 model by using Copy Number Variations (CNVs) and inactivating mutations data on a particular gene target to split the patient dataset in control and inactivated groups. Then, we obtained RNA (RNA-seq) expression levels to extract a gene deletion signature. Here, we show that this method can still be a useful resource as an integrated tool for molecular knowledge mining.

## 2. Implementation

The algorithm is based on the workflow shown in Figure 1: the main inputs of the model are the gene of interest α and the particular tissue context Ʃ (chosen from the available TCGA cohort codes, e.g., ACC and COAD). Next, the inactivated and control sample sets are built. In the first step, only samples for which both CNV and mRNA-seq data are present in the TCGA database are included.

The inactivated sample selection is performed following two different strategies: in the first one, both deletions and inactivating mutations if provided are used to include samples; in the second one, samples are selected only by inactivating mutations.

The deletion-based filter extracts inactivated samples by selecting CNVs that overlap the gene α location and in which the CNV-GISTIC score [12,13] is lower than −1. An optional filter allows to include only deletions larger than a given threshold. The second filter is based on inactivating mutations and requires an input file containing a list of protein substitution variants in the standard format according to Sequence Variant Nomenclature amino_acid/position/new_amino_acid (e.g., Cys28Ser). Unlike the first filter, it incorporates samples with variations present in the inactivating mutation list. Moreover, the specific format “STOP N” can be added to the list, where N is a number representing the rightmost stop-gain mutation allowing a sample to be included in the set.

The control set is built starting from only samples with both CNV and mRNA-seq data. Other exclusion criteria for the control set include outside the gene α, samples containing CNVs larger than a given threshold (e.g., 1 Mb), or mutations inside the same gene α. With these filters, we tried to minimize the genomic interference of having huge structural rearrangements in the control set.

The downstream analysis is executed only if there are at least five samples in the inactivated set and if the ratio between such a set and the control set is higher than a given threshold (0.05).

RNA-seq raw count data are transformed in count per millions (CPM), and only genes for which CPM is greater than 5 in at least 5 samples are kept.

The second part regarding the signature extraction is performed in two separated methods: the first one is a Differential Expression (DE) strategy in order to fetch up- and downregulated genes with regards to the inactivated set. Secondly, a Random Forest approach (RF) is employed with the aim of building activated and inactivated sets from a binary classifier. From the RF, we extracted a gene ranking list that allows to understand the most discriminatory genes in the classification process and the most likely to be part of our signature.

In the DE approach, the voom function of the limma package is executed on the data and a linear model followed by empirical bayesian statistics are performed in order to find differentially expressed genes between the two sets. On the other hand, Random Forests are built via the randomForest function of the randomForest package which implements the Breiman’s random forest algorithm for classification. The preprocessing part is performed via custom Python scripting, whereas the filtered sets are provided as input to an R script that will perform the second step with the voom limma and randomForest [14,15] packages. The data fetch process is automated thanks to the Firebrowse package [16].

The DE output file contains a list of genes with some features, such as the log fold-change and q-value, where the user can observe the putative differentially expressed genes. We appended additional columns to the differential output file in order to give more information on the kind of induction adopted, e.g., two columns with a median expression for each group. The RF output file contains a list of genes ranked by their meanDecreaseGini value, thus having the most important genes in terms of loss of information on top.

The tool is freely available at https://gitlab.com/bioinfo-ire-release/icaro.

## 3. Results

In order to demonstrate the accuracy of our approach, we extracted 50 pairs of frequently deleted genes (and their matching datasets) from the cBioPortal [17] (Appendix A) to run the workflow with. Afterwards, from the output signature, we extracted the fold change and the adjusted *p*-value of the target gene to understand whether we are selecting samples in which the target gene is significantly downregulated. Indeed, almost all of the targets are significantly downregulated (94.0%) and have a strong induction (i.e., log_2_FC < −0.58, meaning a 50% regulation, 93.6%) (Figure 2). We performed a similar benchmarking for the RF results on the same genes. When visualizing the meanDecreaseAccuracy (MDA) and meanDecreaseGini (MDG) of such genes, we observed that only 5/50 (10%) gene-dataset pairs had an MDG higher than 1%, while only 2/50 (4%) pairs showed an MDG over 5% (Appendix A).

We pointed out that inactivated set sizing was the main failure in the workflow. That is, for most datasets, it was difficult to find a high number of patients with focal deletions inside a particular gene. For the RF classification task, it seemed that the deleted gene expression level did not contain a sufficient amount of information in this in vivo setting in order to build a good classifier by itself.

Next, we attempted to demonstrate that the algorithm was able to correlate with other data that were more similar to the typical laboratory approach. The idea involved testing whether the ICAro signature had significant similarities to shRNA knockout perturbations, the routine approach, or other drugs and kinase signatures. To this purpose, we used the aforementioned 50 signatures and we queried L1000 via the Enrichr API [8,9,18] for correlating with the Chemical, Kinase, and Ligand Perturbation. We divided the signatures into up- and downregulated genes; therefore, for each gene-dataset pair, we extracted a L1000 table, 300 in total (Figure 3). On average, every signature correlated with 5 significant terms (adjusted *p*-value < 0.05, median: 5 terms, and mean: 306 terms). When analyzing the particular sub-signatures, up-signatures tended to poorly overlap (median: 0) while down-signatures had better correlation (median from 2 to 660) (Appendix A). This difference is to be clearly attributed to the nature of the model that we tested. In fact, our focus is on deletions; therefore a direct gene downregulation trend will overlap better than an in-trans upregulation event.

In order to show a comparison on the difference between this performance and random distribution, we ran a parallel script, where given $N_{i},\;M_{i}$, the number of significant genes from each signature $S_{i}$, we extracted $N_{i},\;M_{i}$ random genes and executed the Enrichr analysis on them. The median number of significant signatures was 0 (adjusted *p*-value < 0.05, median: 0 terms, and mean: 36 terms), and five out of six classes had a median term number 0 (Figure 3 and Appendix A). The mean amount of terms resulted significantly more in 2 out of 6 cases, particularly in the Chemical Perturbation Down and the Ligand Perturbation Down clusters, confirming the aforementioned hypothesis of the ICAro applicability.

As a second validation process, without focusing on frequently deleted genes, we applied the workflow on the genes of interest in tumor genomics, i.e., cancer driver genes. We focused on 459 mutational cancer driver genes (Appendix A), deriving from the Integrative Onco Genomics (intOgen) list [19]. Among those, we excluded 23 of them, which were located in sexual chromosomes. Given that we did not separate patients by gender, this would have had a strong bias in the CNV/mRNA separation. The analysis was carried out on 35 datasets (Appendix A): only on UCS (Uterine Carcinosarcoma), no results were obtained. For the other datasets, there was a high variability in the number of analyses successfully performed, starting from 2 for CHOL (Cholangiocarcinoma) and DLBC (Diffuse Large B-cell Lymphoma) to 100 for OV (Ovarian serous cystadenocarcinoma), with a mean of 22 successful runs per dataset. From a gene-centered perspective (Appendix A), we obtained at least 1 result from 148 genes (34%) and f in which the minimum is 1 for 60 genes and the maximum is 31 for the WNK1 gene, with a mean of 5 analyses for each gene. The main challenge in performing an ICAro analysis is the lack of CNVs on the genes of interest: 59% of analyses failed for this reason. Subsequently, the second main cause for this failure is the absence or the low amount of inactivated mRNA samples: 88% of samples which had passed the previous filters were rejected at this step. Eight analyses were not performed due to a missing control sample. In conclusion, on the whole, only 5% of analyses were successfully performed.

The final step of ICAro modeling features also a Random Forest analysis in addition to the Differential Expression. The aim is to overcome the limitations of linear modeling and to provide a clean gene rank in terms of importance. In order to further describe the relationship among the two analyses, we compared the results of ICAro executions of the aforementioned 50 gene-datasets pairs in terms of the Differential Expression vs. Random Forest results. This profiling presents different scenarios, in which in some cases, the RF approach can massively extend the scope of the DE, that features only a few significant genes (5 out of the top 100 RF genes are significant in DE, Figure 4A). In other cases, the situation is the opposite, and the RF is only an extension of the strong amount of significant DE genes (75 out of the 100 top RF genes are significant in DE, Figure 4B). The full 50 plots are available at the application’s webpage.

Finally, in order to present the scope and the possible applications of our system, we produced a use case. We exploited a public transcriptomic dataset (GSE76689), a silencing experiment designed to dissect the role of RB1 in Ovarian carcinoma [20]. We reproduced the DE analysis of the paper. Globally, 2 down- and 8 upregulated genes are confirmed to be significant by the system, thus stressing the importance of these mRNAs to discriminate signatures of RB1 loss in Ovarian carcinoma (Table 1). Furthermore, the Random Forest modeling returned 5/10 of the significant genes to be in the top 100 Gini index ranking.

Taken together, these results highlighted that the algorithm is able to extract a few significantly correlated regulation signatures for genes that are frequently deleted in cancer. The workflow performed better than random sampling and could be used by researchers to extract several “parent” signatures from the target gene in a tumor environment. From the cancer gene driver’s point-of-view, a small fraction could be queried for gene signatures thanks to ICAro. Finally, it can be exploited to select a subset of genes of interest in a mRNA profiling experiment.

## 4. Discussion

The intricate patterns of transcriptional networks are complex to decipher for the biomedical researcher, and in our experience, researchers struggle to find evidence to confirm a regulatory hypothesis. This is one of the main reasons that led us to develop a simple algorithm to help investigators in the field of Cancer Transcriptomics.

The other motivation comes from our experience in handling NGS data and bioinformatic analysis of a medium-sized genomic facility. Translational projects are often designed to start with a whole transcriptomic or a whole epigenomic experiment (e.g., RNA-seq and ChIP-seq), intended to be the hypothesis driver for further investigations. As a matter of fact, the process risks to be interrupted when bioinformaticians present researchers with enormous lists of genes and ontologies. We impute this matter to three main factors: the lack of computational biologists in research groups, the intrinsic difficulty of summarizing large quantity of data, and a slow validation process due to the high number of possible targets as starting points. ICAro comes as an aid for the latter issues, providing hints on mRNA targets that could indeed be validated in vivo.

Many confounding factors are not taken into account in the patient partitioning. These are, for instance, patient stratification by demographic data. This is an issue of many algorithmic signatures of the transcriptomic field that do not seem to care even if they are designed to stratify patients into clinical settings [21,22]. In our case, the scarcity of the inactivated set, usually falling below the count of 5, prevents us in further dividing the patient strata.

In addition, most TCGA mutation datasets do not carry Variant Allele Frequency (VAF) information. For this reason, we may erroneously include a few patients in the inactivation set (that is already suffering from typical smaller size) that carry a stop-gain mutation in only a small fraction of tumor cells (e.g., VAF < 10%). This limitation also applies to CNV data, where the GISTIC threshold output are decided on a sample by sample basis [23]. Furthermore, it should be noted that every sample profiled in the TCGA had a tumor cellularity of at least 80% (recently shifted to 60%) and is not available metadata for which we could correct the CNV/Mutation status.

Our implementation process also lacks some features that we plan to employ in the future. The most obvious one is the lack of a gene amplification study. That is, the possibility to extract a signature when a gene has more copies. This could be a valuable experiment mirroring another frequent laboratory approach such as overexpression models. Another interesting add-on would be appending genomic coordinates of each gene locus to the final output in order to understand whether the differential effect is mostly guided by the CNV itself or by some other regulation pathways. Finally, one more aspect that could be improved in the future is the simple automatization of functional APIs from the result dataset, such as LINCs Cloud and ENRICHR, allowing researchers to better investigate the mechanisms involved.

ICAro testing on a list of mutational cancer driver genes pointed out that the main problem is that less than half of such genes are affected by CNVs, and among the samples with these deletions, only 1 over 12 contains related mRNA experiments, thus preventing us from performing the analysis on a larger set of data.

## 5. Conclusions

Mining knowledge regarding gene function or seeking inactivation biomarkers is not so trivial tasks. It is for this reason, we developed an automated tool to integrate and mine knowledge from third-level TCGA data. Our testing showed that this workflow is able to extract several transcriptional signatures for a discrete set of genes.

From a biological perspective, the authors are aware that (a) the amount of patients with focal deletions for a given gene will be discrete for the time being, (b) the cancer genomic and transcriptomic background is a disorderly environment very different from engineered cell lines, and (c) it is known that most frequent gene losses have recurrent breakpoints [12]. Nevertheless, we remain confident in the value and feasibility of the presented approach due to the rapid increase in the amount of available high-throughput data and in the vast disappointing failures of in vitro derived models.

We are currently working on an extended version for miRNA signature extraction that will be useful for researchers in the non-coding RNA field. Investigators will fetch via ICAro differential miRNA classes that are up-and downregulated by a particular gene deletion, providing additional insights on miRNA-mRNA interaction.

In a real-life setting, we trust that the ICAro approach would be of value when paired with several other approaches such as in vitro or in vivo knockout models, for instance when understanding biomarkers for the inactivation of a particular gene. In this scenario, it will be useful to implement a novel branch of the workflow to take into account also other emerging large-scale omic approaches such as Reverse-Phase Protein Arrays (RPPA).

## Figures and Tables

**Figure 1 cancers-11-00256-f001:**
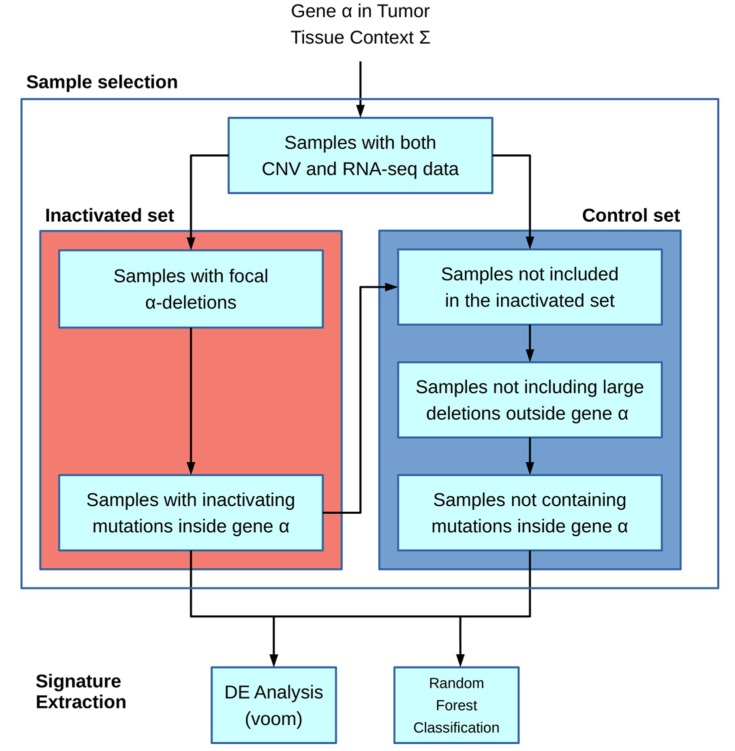
A schematic representation of the ICAro (gene signature Inference system from Copy number Aberrations) workflow. The first part relies on sample filtering based on deletions and inactivating mutations spanning the gene target α in order to build the inactivated and the control sample sets. They are used as input for the signature extraction process, performed via a differential gene expression analysis (the voom function from limma) and a Random Forest classification (randomForest R package).

**Figure 2 cancers-11-00256-f002:**
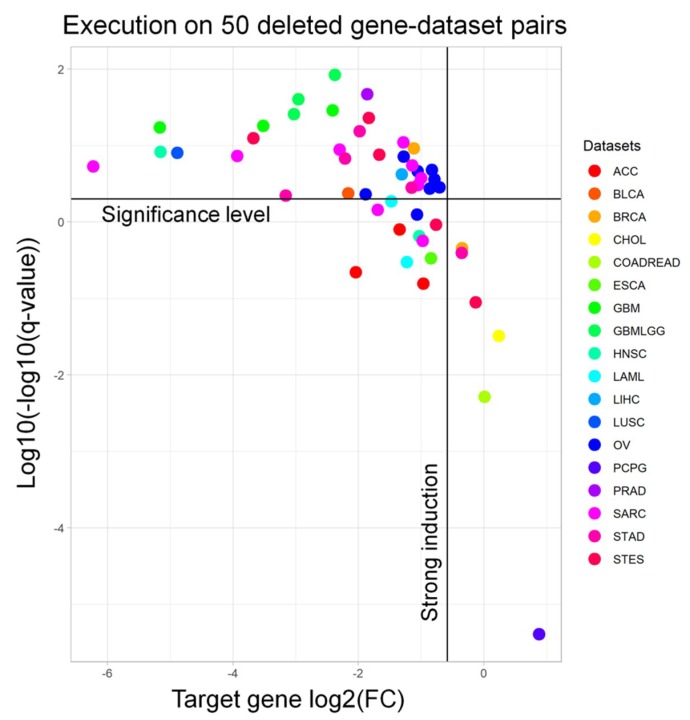
The performance of the ICAro Differential Expression for 50 executions on frequently deleted gene-dataset pairs: Every point represents one ICAro execution on a gene-dataset pair (e.g., TP53 on COADREAD). The different colors represent several TCGA datasets. *X*-axis: log2FC (gene induction), *Y*-axis: transformed *q*-value (statistical significance). Most tests fall in the upper region, meaning that they are significant, and on the center of the *X*-axis, i.e., they are downregulated. The downregulation of deleted genes is a first step towards the in vivo validation of the ICAro process. For a complete key of datasets please refer to Appendix A.

**Figure 3 cancers-11-00256-f003:**
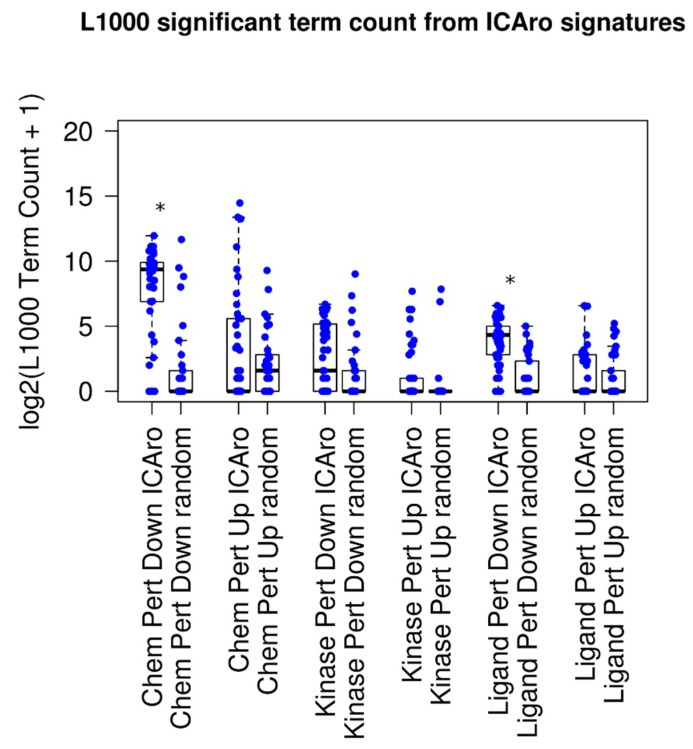
The amount of significant terms for down and up-regulated genes when compared to L1000 signatures: Every point is an ICAro execution with a significant gene set (up or down). Every signature is compared with the amount of significant terms when sampling random gene sets of equal size. Legend: * significant increase between random sampling and ICAro.

**Figure 4 cancers-11-00256-f004:**
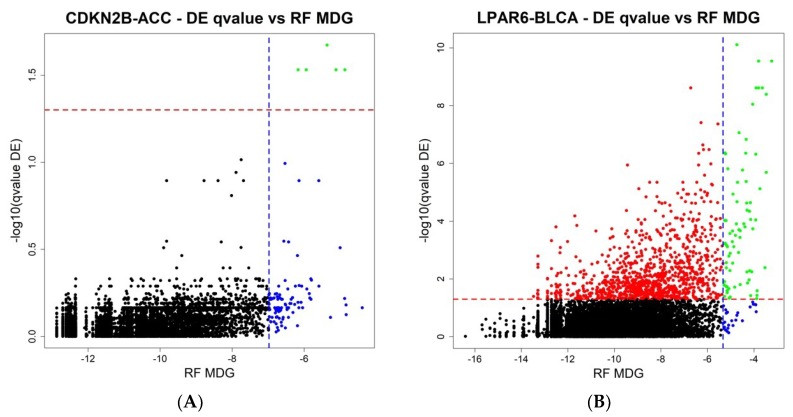
(**A**,**B**) Two representative plots of the Differential Expression against the Random Forest analysis on the ICAro system. Blue line: the top 100 genes from the Random Forest analysis, ranked by the meanDecreaseGini (MDG). Red line: the adjusted *p*-value significance threshold. Left: only a few genes are significantly regulated in the DE analysis, but more can be studied from the top 100 genes on the RF analysis. Right: the opposite situation where most information lies in the differential expression, and just most of the top 100 RF genes are significant in DE terms.

**Table 1 cancers-11-00256-t001:** The significant genes validated in the GSE76689 dataset from the ICAro system.

Gene	Log2FC siRB1	Log2FC ICAro	adj PVal siRB1	adj P Val ICAro
RB1	−0.83	−1.12	5.73 × 10^−4^	1.81 × 10^−6^
SH3BP4	−0.64	−0.63	8.21 × 10^−4^	3.35 × 10^−2^
NUDT21	0.67	0.29	1.22 × 10^−3^	4.46 × 10^−2^
SLC27A3	0.64	0.40	5.70 × 10^−3^	3.45 × 10^−2^
C15orf38	0.77	0.42	1.40 × 10^−3^	3.82 × 10^−2^
ADCY3	0.76	0.49	7.18 × 10^−4^	1.78 × 10^−2^
TMEM106C	0.66	0.51	2.69 × 10^−3^	2.70 × 10^−2^
FANCE	0.47	0.57	2.92 × 10^−2^	2.36 × 10^−3^
WDR34	0.53	0.59	1.31 × 10^−2^	4.94 × 10^−4^
TCF19	0.49	0.99	2.68 × 10^−2^	1.81 × 10^−6^

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
