# Peer review of "Reverse Engineering Cancer: Inferring Transcriptional Gene Signatures from Copy Number Aberrations with ICAro"

_cancers, 2019, doi:10.3390/cancers11020256_

Reviewer 1 Report

The authors present a new tool able to mimic in silico CRISPR/CAS experiment.

As stated by the authors, methods required improved to gain more biological insight from publicly available dataset.

At this exploratory level, the tool bioinformatic tool is interesting and could help researcher in data mining and to support experimental results.

Minor revisions:

- improve description of Figure 2;

- will the authors include clinical info in the selection criteria of the cohort?

Author Response

############ Reviewer #1 #################

The authors present a new tool able to mimic in silico CRISPR/CAS experiment.

As stated by the authors, methods required improved to gain more biological insight from publicly available dataset.

At this exploratory level, the tool bioinformatic tool is interesting and could help researcher in data mining and to support experimental results.

Minor revisions:

- improve description of Figure 2;

- will the authors include clinical info in the selection criteria of the cohort?

Response

We thank the reviewer for the insights concerning our work.

We did extend the legend description of Figure 2.

On the clinical info extension, we tried to reason on how to to include it in the selection criteria. One possibility would be selecting only patients with a certain level of OS/PFS, thus selecting only top/bottom survivors. If the reviewer refers to other clinical data annotation from TCGA, e.g. Colorectal Cancer classes (e.g. MSI high/low), unfortunately they are strongly variable among cancer types and they cannot be generalized with our “pan-cancer” workflow. Another issue is that we would make the inactivated set even smaller, and this is already a limiting factor of our workflow.

However, these annotations can be included by any bioinformatician in order to make the selection more “fine-grained”. In order to perform a cancer specific clinical/metadata partitioning of the dataset, our gitlab available workflow is highly customizable. The Firebrowse API (i.e. Sample/Clinical subset) can be exploited to this purpose. We did add comments on the README section of the gitlab on this matter.

Reviewer 2 Report

Review of "Reverse engineering cancer: inferring transcriptional gene signatures from copy number aberrations with ICAro" by D. Angeli, M. Fanciulli and M. Pallocca.

General comments:

The authors deal with the problems related to the translation process from in vitro characterization results to human validation and propose a "mRNA signature" generated by a workflow called "ICAro". The signature is claimed to be related to a gene loss event thus helping the understanding of its function and, possibly, the possibility of its usage as a reliable gene-loss biomarker. As a validation set, Angeli et al. use a 50-gene datasets couples from th cBioPortal. Moreover they used 459 mutational cancer driver genes obtained from intOgen. The results are quinte interesting and very promising since the algorithm is able to extract significantly correlated regulation signature for frequently deleted genes, also in cancer.

The authors must be complimented for a very accurate and clearly explained work. I particularly liked the depth of statistical reasoning that was expressed as a 'procedural flux'in a step-by-step way linking the methodological choices to the biological problems. This is exactly how we must proceed in empirical research without claiming for 'ready made' magic algorithms.

However, my concern - and hence request for the authors - is the following: as is often the case, a lot of methods paper fail to clearly identify what precisely was achieved in terms of the fundamental question: at the end of the day, what has been learned using this method or what specifically has been achieved that would have been otherwise impossible. In other words I wish the authors provide some concrete discussion as to what the biological implications of the method is. My concern is that many outstanding methods, like the one the authors present, are eventually lost because it is not clear by reading the paper, how the method would help another scientist. So, I think that it would add tremendous value to the manuscript if the authors could elaborate more on the biological interpretation and implications of their findings, compared to other approaches.

Author Response

############ Reviewer #2 #################

General comments:

The authors deal with the problems related to the translation process from in vitro characterization results to human validation and propose a "mRNA signature" generated by a workflow called "ICAro". The signature is claimed to be related to a gene loss event thus helping the understanding of its function and, possibly, the possibility of its usage as a reliable gene-loss biomarker. As a validation set, Angeli et al. use a 50-gene datasets couples from th cBioPortal. Moreover they used 459 mutational cancer driver genes obtained from intOgen. The results are quinte interesting and very promising since the algorithm is able to extract significantly correlated regulation signature for frequently deleted genes, also in cancer.

The authors must be complimented for a very accurate and clearly explained work. I particularly liked the depth of statistical reasoning that was expressed as a 'procedural flux'in a step-by-step way linking the methodological choices to the biological problems. This is exactly how we must proceed in empirical research without claiming for 'ready made' magic algorithms.

However, my concern - and hence request for the authors - is the following: as is often the case, a lot of methods paper fail to clearly identify what precisely was achieved in terms of the fundamental question: at the end of the day, what has been learned using this method or what specifically has been achieved that would have been otherwise impossible. In other words I wish the authors provide some concrete discussion as to what the biological implications of the method is. My concern is that many outstanding methods, like the one the authors present, are eventually lost because it is not clear by reading the paper, how the method would help another scientist. So, I think that it would add tremendous value to the manuscript if the authors could elaborate more on the biological interpretation and implications of their findings, compared to other approaches.

Response

We thank the reviewer for the insights concerning our work.

Regarding the biological takeaways, we added two new sections to the manuscript.

First, we conducted a comparison between the Differential Expression and the Random Forest approach, showing their complementarity, on the 50 gene-dataset couples used in the validation. This data adds to the whole picture an evidence that by using “orthogonal” techniques it can be possible to discover different information. 

Secondly, we provided a “Use Case” in which we compare a published transcriptomic dataset (Comisso et. Al, Oncogene 2017) produced from a gene silencing, and we cross-compared the results with the ICAro signature on the same cancer dataset (Ovarian). The intersections shows a handful of genes (10) from the ICAro signature that overlap the differential expression (2 down, 8 up). These can be further investigated in a scenario of a RB1-loss related research, or to further understand the RB1-ome network in Ovarian Cancer patients.

Finally, we further specified in the discussion section the motivation and the biological motives behind ICAro implementation.

Reviewer 3 Report

A very elementary analysis is performed by separating samples using CNV information and performing differential expression analysis on the transcriptomics data. The authors use TCGA data and define case group as the one with focal deletions and inactivating mutations in a gene. Such an analysis is useful only as a first step to explore the variation in the dataset and cannot be an end unto itself. If you separate samples into groups and perform linear modeling (without correcting for any covariates or confounding factors) you will find some gene signatures. The analysis provided by the authors is neither rigorous nor any meaningful new biology can be extracted from it. Both the main analyses including the comparison of ICAro (no full form given when it occurred first) to shRNA-knockout data from L1000 and the one focusing on cancer driver genes, fail to provide me any key takeaways from the results. ICARO (multiple version of capitalizations) finds down-regulated signatures have better overlap since the authors are focusing on deletions is a truism and sounds more matter-of-fact than a strong result. In cancer genomics, the authors mention globally only 5% of analyses were successfully performed.

The authors indeed mention extensions of ICAro, what additional biology can be uncovered using ICAro but that hardly adds any merits to the current results obtained by ICAro. I think the paper does not fit the scope of the journal MDPI-Cancers and maybe suitable elsewhere upon serious revision to improve readability and the scientific import/message.

Author Response

############ Reviewer #3 #################

A very elementary analysis is performed by separating samples using CNV information and performing differential expression analysis on the transcriptomics data. The authors use TCGA data and define case group as the one with focal deletions and inactivating mutations in a gene. Such an analysis is useful only as a first step to explore the variation in the dataset and cannot be an end unto itself. If you separate samples into groups and perform linear modeling (without correcting for any covariates or confounding factors) you will find some gene signatures. The analysis provided by the authors is neither rigorous nor any meaningful new biology can be extracted from it.

We thank the reviewer for all the comments regarding our work.

We acknowledge the limitations of simple linear modeling of data without taking account the many confounding factors that genomic-only modeling carries by itself. For this reason, we also included a Random Forest approach, that is a non-linear model able to extract information in more complex data distributions.

Furthermore, we addressed a few of the other confounding factors that are specific of genomic samples themselves, i.e. large genomic alterations and mutations on the same target gene. Of course, other criteria are lacking in order to make the results cleaner, e.g. stratification of samples per patient’s age, tumor purity, etc. Unfortunately, the scarce cardinality of the “inactivated set” (not many patients carry focal deletions for a gene) prevents us to further normalize the sampling.  We did add comments on limiting scenario in the discussion of the manuscript.

In order to better present the interaction between the two analytical approaches (Differential Expression and Random Forests) we also did produce a series of plots/comparison for each of the 50 gene-datasets couples used for the validation, by plotting random forest ranking (gene importance) vs differential expression significance (qvalue). The results are presented in Fig. 4A-B, full plots available on https://gitlab.com/bioinfo-ire-release/icaro. The takeaway message is that in many cases the RF modeling can extend the DE results.

Both the main analyses including the comparison of ICAro (no full form given when it occurred first) to shRNA-knockout data from L1000 and the one focusing on cancer driver genes, fail to provide me any key takeaways from the results. ICARO (multiple version of capitalizations) finds down-regulated signatures have better overlap since the authors are focusing on deletions is a truism and sounds more matter-of-fact than a strong result. In cancer genomics, the authors mention globally only 5% of analyses were successfully performed.

We fixed the capitalization versions of the system name and provided the full form at the beginning of the introduction.

Regarding down regulation: we agree that the down-regulation of the target gene is just a validation. But the downregulation signatures in the L1000 have many compounds tested for mRNA signatures, not only shRNAs.

The authors indeed mention extensions of ICAro, what additional biology can be uncovered using ICAro but that hardly adds any merits to the current results obtained by ICAro. I think the paper does not fit the scope of the journal MDPI-Cancers and maybe suitable elsewhere upon serious revision to improve readability and the scientific import/message.

In order to make clearer a possible ICAro application, we produced a Use Case in which we compared a real-life published transcriptomic dataset (Comisso et. Al, Oncogene 2017) produced from a gene silencing, and we cross-compared the results with the ICAro signature on the same cancer dataset (Ovarian). The intersection shows a handful of genes (10, 2 down, 8 up) that are significant with the same fold change direction. These genes can be selected for further validation with other techniques. In this context, as we specified in the extended discussion, ICAro can be a useful tool in order to reduce the investigation space for molecular biologists that are usually presented with huge tables containing thousands of genes, for instance coming from an RNA-seq experiment. In our experience, many projects fail or are strongly slowed by the lack of direction in these amount of data. 

Regarding readability, the whole manuscript has been revised by an English mother-tongue proof reader. 

Round  2

Reviewer 3 Report

The authors have addressed my concerns and have improved the quality of the manuscript. 

few suggestions,

1) Improve figure 2,4, change the background, increase font size, either change the color or change the symbol to show cancer type. I vote change color by keeping pch=19.

2) Not necessary but just a suggestion, replace gene-dataset couple by gene-dataset pair.

3) Avoid words like unfortunately, ambitious. These are not-scientific words and they hinder objectivity.

4) So many minor typos, please read carefully.

 line 78, 

Here, ewe show that this method can still an useful resource -> Here, we show that this method can still be a useful resource

Author Response

Reviewer # 3

The authors have addressed my concerns and have improved the quality of the manuscript. 

few suggestions.

We thank the reviewer for the suggestion regarding the quality and readability of our manuscript.

1)    Improve figure 2,4, change the background, increase font size, either change the color or change the symbol to show cancer type. I vote change color by keeping pch=19.

We changed Figure 2 according to the reviewer’s suggestion. We increased font size of figure 4 to increase readability.

2)    Not necessary but just a suggestion, replace gene-dataset couple by gene-dataset pair.

We did replace all “couple” occurrences with gene-dataset pair, more mathematically sound.

3)    Avoid words like unfortunately, ambitious. These are not-scientific words and they hinder objectivity.

We removed the “colloquial tone” words and replaced them with more formal ones, especially in the introduction.

4)    So many minor typos, please read carefully.

We worked through all the manuscript and fixed a few more typos.